# Patient-reported outcome measures for acne: a mixed-methods validation study (acne PROMs)

Samantha Hornsey ,[1] Beth Stuart,[1] Ingrid Muller,[1] Alison M Layton,[2,3] Leanne Morrison,[1,4] Jamie King,[1] Karen Thomas,[5] Paul Little ,[1] Miriam Santer[1]

SH and BS are joint first authors.

[1]Faculty of Medicine, Primary Care, Population Sciences and Medical Education, University of Southampton, Southampton, UK
[2]Department of Dermatology, Harrogate and District NHS Foundation Trust, Harrogate, UK
[3]School of Medicine, Hull York Medical School, York, UK
[4]Department of Psychology, University of Southampton, Southampton, UK
[5]Faculty of Medicine, Primary Care, Population Sciences and Medical Education, PPI Representative, University of Southampton, Southampton, UK

**Correspondence to**
Samantha Hornsey;
sh7g13@soton.ac.uk

## ABSTRACT

**Objectives** To examine the acceptability and validity of two patient-reported outcome measures (PROMs) for adult acne, comparing them to the validated Acne-specific Quality of Life (Acne-QoL) measure.

**Design** Mixed-methods validation study.

**Setting** Participants were recruited by (1) mail-out through primary care if they had ever consulted for acne and received a prescription for acne treatment within the last 6 months, (2) opportunistically in secondary care and (3) poster advertisement in community venues.

**Participants** 221 (204 quantitative and 17 qualitative) participants with acne, aged 18–50 years.

**Outcome measures** Quantitative sub-study participants completed Acne-QoL, Skindex-16 and Comprehensive Acne Quality of Life Scale (CompAQ) at baseline, 24 hours and 6 weeks. Qualitative sub-study participants took part in cognitive think-aloud interviews, while completing the same measures. Transcribed audio recordings were analysed using inductive thematic analysis.

**Results** Quantitative analyses suggested high internal consistency (Cronbach's alpha 0.74–0.96) and reliability (intraclass correlation coefficient values 0.88–0.97) for both questionnaires. Both scales showed floor effects on some subdomains. Skindex-16 and CompAQ showed good evidence of construct validity when compared with Acne-QoL with Spearman's correlation coefficients 0.54–0.81, and good repeatability over 24 hours.

Qualitative data uncovered wide-ranging views regarding usability and acceptability. Interviewees held strong but differing views about layout, question/response wording, redundant/similar questions and guidance notes. Similarly, interviewees differed in perceptions of acceptability of the different scales, particularly on relatability of questions and emotive reactions to scales.

**Conclusions** All PROMs performed well in statistical analyses. No PROM showed superior usability and acceptability in the qualitative study. Any PROM should be acceptable for further research in adult acne but researchers should consider the different domains and whether they will measure only facial or facial and trunk acne before making a selection. A new PROM or further evaluation of novel PROMs may be beneficial.

## INTRODUCTION

Acne vulgaris is a very common condition among adolescents and adults. Some degree of acne affects almost all people aged 15–17

### Strengths and limitations of this study

► Triangulated findings through a mixed-methods approach.
► Sufficient sample size for both the qualitative and quantitative samples, in line with Consensus-based Standards for the Selection of Health Measurement Instruments guidelines.
► Participants in the quantitative study had a wide range of acne severities and varied experiences of different treatments (predominantly female, which may not represent the true population with adult acne).
► No interviewees were recruited through secondary care and most survey participants were recruited through primary care, which may limit generalisability.
► Acne severity was self-reported.

years and is moderate to severe in about 15%–20% of people, often persisting to adulthood.[1] Clinically relevant facial scarring occurs in approximately 20% of cases. Both acne and acne scarring can impact negatively on psychosocial dimensions and psychological disability may be equivalent to that seen in chronic conditions such as asthma and diabetes.[2] Negative impact on quality of life (QoL), with increased risk of depression and suicide has been reported.[1 2]

Over 3% of people aged 13–25 years consult for acne each year.[3] Although topical treatments, such as benzoyl peroxide and topical retinoids, can be effective, there is uncertainty regarding the most appropriate strategy for initial and maintenance treatment.[1] Current National Institute for Health and Care Excellence UK Guidelines suggest first-line treatment for mild to mild/moderate disease is single agent topical, followed by combination topical treatment with or without oral antibiotics.[2] International guidelines differ, suggesting combination topical regimes, rather than single agent topical treatment should be used first line for mild-to-moderate

disease.[4–6] Further research is needed to clarify which topical treatment and/or oral treatments are most effective in mild-to-moderate acne.

Trials of acne treatments have used a wide range of different outcome measures, hampering interpretation of research findings, particularly secondary research and direct comparisons of treatments.[1 7] However, in order to determine which treatment is most effective in clinical trials, researchers need to be able to reliably measure and compare treatment response between trials. Therefore, the lack of consensus over suitable outcomes needs to be resolved in order to carry out robust future trials and systematic reviews.

### Acne outcomes

The Acne Core Outcomes Research Network (ACORN) is an international group seeking to develop a core set of outcome measures for acne because, 'The lack of standardisation of how acne is assessed in clinical trials makes it challenging to pool data from different trials of the same treatment and impossible for clinicians to know the relative effectiveness of different types of treatment.'[8] However, this process is in development and to date has focused on developing consensus on core outcome domains rather than developing new measures or further validating existing measures.[9] In the meantime, the US Food and Drug Administration (FDA) has recently updated guidance suggesting that all trials should include lesion count and an Investigator Global Assessment (IGA) with no mention of patient-reported outcomes.[10]

Lesion counts are time-consuming, with wide interassessor variation and give little additional information to global assessments.[11] The FDA advocates the use of a 5-point global assessment ranging from clear to severe. However, global severity scales, such as the IGA and Comprehensive Acne Severity Scale, struggle to show sensitivity to change, which can be crucial in a clinical trial setting.[12 13]

Although outcomes in dermatology have traditionally been measured by 'objective' or clinician assessment, there is increasing recognition that the patient's experience of the condition may be a more relevant outcome measure, particularly in pragmatic or 'real world' trials.[14] As acne is frequently a long-term condition with psychological impact, a patient-reported outcome measure (PROM) that can be used at repeated time points over the duration of a trial would be an attractive outcome measure.

### AIMS

This study aims to explore the acceptability and validity of two potential PROMs using a mixed-methods approach. The results will inform choice of outcome measures for future trials and potentially support the ACORN core outcomes initiative.

### METHODS
### Design

This was a mixed-methods study exploring the acceptability and validity of two disease-specific acne instruments, the Acne-Specific Quality of Life Questionnaire (Acne-Qol)[15–17] and the Comprehensive Acne Quality of Life Scale (CompAQ),[18] along with a general skin instrument, the Skindex-16.[19–21] The validity of CompAQ[18] and Skindex-16[19–21] was compared with published results for the Acne-QoL. Cognitive think-aloud interviews[22] for all instruments were also conducted.

### Outcome measures

The outcome measures in this study (see table 1) were selected after discussion with a panel of researchers, dermatologists, general practitioners and patients to ensure that they represented the most suitable instruments for further study.

### Participants/data collection

Data were collected between August 2017 and June 2018. People were eligible to take part in the study if they were aged 18–50 years and had acne. Participants were recruited through primary care, secondary care and community advertising. In primary care, database searches identified potential participants who had consulted with acne and been prescribed an acne treatment in the past 6 months. Lists were screened and people were excluded prior to postal mail-out if they had severe distress, a known opposition to participating in research or known not to have a good understanding of written English.

Secondary care participants were identified opportunistically from acne clinics in three NHS Trusts. For community advertising, advertisements were placed around a university campus with study contact details. People who saw the advert and were interested in taking part emailed to express an interest and the researcher phoned them to confirm eligibility before sending them the study pack or arrange an interview.

Participants were invited to take part in either the qualitative or quantitative study. Qualitative interviewees were given a £10 voucher to thank them for their time and survey participants were entered into a prize draw to win a £100 voucher.

### Sample size

There is no agreed basis for calculating a sample size for a validation study. However, Consensus-based Standards for the Selection of Health Measurement Instruments (COSMIN) guidelines suggest that studies can be considered 'excellent' if the sample size is over 100 participants.[23] It has been suggested that 10 participants per questionnaire item is sufficient[24] or that there should be at least three times as many participants as there are questionnaire items.[25] There are 16 items in the Skindex-16, 20 items in the CompAQ and 19 items in the Acne-QoL questionnaires. Therefore, we aimed to recruit 200 participants to the quantitative study.

The qualitative study aimed to include 15–20 participants where data saturation was expected. Purposive sampling was used to select study participants to ensure a wide range of views and experiences were represented.

**Table 1** Outcome measures selected

| Outcome measure | Number of total items (scale) | Domains (number of items within each domain) | Description |
|---|---|---|---|
| Acne-QoL[15–17] | 19 (7 point) | Symptoms (5) Role emotional (5) Role social (4) Self-perception (5) | Of available PROMs, the Acne-QoL is the most extensively validated for facial acne. The questions ask respondents to score their answer based on the last week. For each item, the answers range on a 7-point scale. The scale points are labelled from 'extremely' to 'not at all', except three of the symptoms subscale items responses which are labelled from 'extensive' to 'none'. The higher the score, the better the respondent's QoL. |
| Skindex-16[19–21] | 16 (7 point) | Symptoms (4), Emotions (7), Functioning (5) | Skindex was designed to be used in a range of skin conditions and is not acne specific, although it has been used in published acne studies. It covers similar domains to the Acne-QoL, but with more focus on appearance. The 29-item Skindex[21] has been more extensively validated than the 16-item version, which has not been validated specifically for acne. However, the shorter version would be preferable in a clinical trial setting, where questionnaire burden can be a concern for participants. In the Skindex-16, the questions ask the respondent how bothered they have been by the items mentioned in the last 7 days. The scale points are labelled with 'Never bothered' for 0 and 'always bothered' for 6 for each item. The scores are converted to a linear scale ranging from 0 to 100; the higher the scores, the poorer the respondent's QoL. |
| CompAQ[18] | 20 (9 point) | Symptoms (4) Social (judgement by others) (4) Social interactions (4) Psychological/ emotional (4) Treatment concerns (4) | The CompAQ is a recently developed outcome measure and, unlike the other two measures, can be used for both facial and truncal acne. The CompAQ was developed in parallel with the ACORN initiative to secure consensus about core outcome domains and has not yet been externally validated. For each item, the respondents are asked to choose how much the item relates to them. The scale points are labelled on every other number with 0 as 'never' and 8 as 'all the time' and therefore, the higher the score, the poorer the respondents QoL. |
| Patient Global Assessment and the Patient Global Assessment of Change | (6 point) | | We also included a patient global assessment of their acne and, at 6 weeks, a patient global assessment of the change in their acne to assist in the assessment of sensitivity to change. The questions used were 'How would you describe your acne at the moment', with responses ranging from clear to very severe, and 'How would you describe the change in your acne in the last 6 weeks?', with responses ranging from 'Completely cleared' to 'Worse'. Participants were encouraged to take a photo at baseline to refer to at the 6-week follow-up to help them answer this question. |

Acne-QoL, Acne-specific Quality of Life; ACORN, Acne Core Outcomes Research Network; CompAQ, Comprehensive Acne Quality of Life Scale; PROMs, patient-reported outcome measures; QoL, quality of life.

For example, trying to include more male participants towards the end of the interview phase, because the sample consisted of more females.

### Quantitative validation
#### Data collection
Participants were invited to complete the three PROMs at baseline, 24 hours and 6 weeks. A study invitation pack, containing information sheet, baseline and 24 hours questionnaires, and a freepost envelope was sent to the patient or handed to them in clinic. Participants wishing to take part completed and returned the questionnaires directly to the researchers by using the freepost envelope. Six weeks following the date of the baseline booklet, the research team sent the 6-week follow-up booklet containing PROMs and questions about acne treatment and severity.

### Statistical analysis
Analysis was undertaken in Stata V.14. The assessment of validity was guided by the COSMIN guidelines and thresholds for the analyses set with reference to the latest quality criteria.[23 26] The copyright holder of the Acne-QoL permitted only the published values to be compared with the other two instruments with the exception of responsiveness to change, which we were given permission to explore in our sample.

#### Internal consistency
Internal consistency was assessed via Cronbach's alpha at baseline, with a good internal consistency being represented by 0.70 and above.

#### Construct validity
Construct validity indicates how much the scale measures what it is intended to measure, and this was

assessed by Spearman's rank correlation coefficient. As there are no 'gold standard' PROMs, the Skindex and the CompAQ were compared with a patient-reported global bother measure and to Acne-QoL. Guidelines suggest that a correlation coefficient of 0.9–1.0 is very high, 0.7–0.9 is high, 0.5–0.7 is moderate, 0.3–0.5 is low and below 0.3 is negligible.[27]

### Reliability

Reliability is the extent to which the scale is free from error and can be tested by asking participants to complete the measures twice, 24 hours apart and calculating the intraclass correlation coefficient (ICC). An ICC of 0.70 and above suggests a good reliability.

### Responsiveness to change

This is the ability of the measurement to detect change over time. It was anticipated that participants' acne would change over a 6-week period and therefore a PROM should be able to detect this change. We hypothesised that for most participants, their global assessment of change would correlate with their change in their PROM scores between baseline and 6 weeks. While we would normally expect a moderate correlation (0.5–0.7), given that this population did not have any intervention and therefore change may not be expected for some participants, we would hypothesise a somewhat lower level of correlation (0.4–0.6).

We also created a binary measure—improved/not improved—based on the participant-reported change and examined the area under the receiver operating characteristic curve (AUROC) for the change in each scale. An AUROC of 0.70 or higher was considered acceptable.

We hypothesised that a change in acne would be more likely in those participants who had experienced a change in treatment over the 6-week period. We therefore also explored these values in this subgroup.

### Interpretability

Interpretability was measured by floor and ceiling effects, defined as more than 15% of the sample achieving the highest or lowest scores.[28]

## Qualitative validation study

### Data collection

Potential participants were invited through primary care mail-out, to take part in the qualitative study. Mail-outs were completed by practices that did not mail-out quantitative invitation packs. Database search and mail-out followed the same procedure as above. Invitation packs included a reply slip for return to the research team if potential participants wished to be contacted about taking part. The researcher (SH) phoned people who returned reply slips to discuss and arrange an interview if they wished to continue. Written informed consent was obtained prior to interview.

Cognitive think-aloud interviews[22] were carried out by SH to explore the acceptability and face validity of the PROMs. During the interview, participants were asked to complete the three PROMs while saying out loud all their thoughts and decision processes. The PROMs were given to the participants in random sequence to avoid questionnaire order effects. An interview topic guide included prompts to use during the interview. It also included semistructured questions about the PROMs that were asked after completing questionnaires. The interviews were carried out face to face, at the participants' home or at the University of Southampton. Interviews were digitally recorded and transcribed verbatim. Interviews were conducted until data saturation was achieved for main themes.

### Analysis

Transcript data were analysed using inductive thematic analysis.[29] A coding schedule was derived by SH by reading, rereading and immersing herself in the data. This schedule was refined throughout the analysis process by discussion with IM and MS, while keeping an audit trail. Codes were applied to sections of the text using NVivo software (V.11).

## Patient and public involvement

Choice of outcome measures was informed by consideration and awareness of the domains that were identified as most important to patients taking part in the James Lind Alliance Priority Setting Partnership for acne.[30 31] Coapplicant and patient and public involvement (PPI) representative, KT, helped with the design of the study and commented on study materials. In addition, an acne PPI panel was formed through advertising on the INVOLVE People in Research portal.[32] Due to geographical constraints, this was a 'virtual panel' who commented by email and by phone. One member of the panel took part in a practice interview over Skype to see if this would be feasible for data collection in cognitive interviews. From this, it was decided that Skype was not feasible. Six members of the panel also gave feedback on initial findings, confirming their relevance.

## Ethical approval

This study was reviewed by the Office for Research Ethics Committees Northern Ireland (17/NI/0054), the Health Research Authority (IRAS ref 219692) and the University of Southampton ethics committee (24489).

## RESULTS

## Participant characteristics

A total of 221 participants were recruited: 204 to the quantitative survey and 17 to qualitative interviews. At baseline and 24 hours, 204 participants completed all the PROMs and at 6 weeks, 167 participants completed the PROMs. No questionnaires were returned with uncompleted measure. All participants who returned the booklets to us completed all the question items on all the included PROMs.

Participants in the quantitative survey were mostly female (85.3%) with a mean age of 28.2 (SD 8.5). Most participants were recruited through primary care and 84.8% had acne for over 2 years. Participants reported having tried a variety of treatments and a number of

**Table 2** Participant baseline characteristics (quantitative study)

| | N (%) or mean (SD) |
|---|---|
| Recruitment site | |
| Primary care | 176 (86.3%) |
| Secondary care | 27 (13.2%) |
| Community advertising | 1 (0.5%) |
| Female | 174 (85.3%) |
| Male | 30 (14.7%) |
| Age | 28.2 (8.5) |
| Duration of acne | |
| Less than 6 months | 4 (2.0%) |
| 6 months–1 year | 9 (4.4%) |
| 1–2 years | 18 (8.8%) |
| 2+ years | 173 (84.8%) |
| Treatments tried ever | |
| None/not sure | 3 (1.5%) |
| Benzoyl peroxide cream/lotion/gel | 118 (57.8%) |
| Topical retinoid cream/lotion/gel | 53 (26.0%) |
| Topical adapalene cream/lotion/gel | 60 (29.4%) |
| Antibiotic cream/lotion/gel | 93 (45.6%) |
| Combination cream/lotion/gel—combining above products | 84 (41.2%) |
| Antibiotics by mouth | 163 (79.9%) |
| Contraceptive pill | 113 (55.4%) |
| Cocyprindiol by mouth | 43 (21.1%) |
| Isotretinoin by mouth | 53 (26.0%) |
| Treatments currently using | |
| None/not sure | 19 (9.5%) |
| Benzoyl peroxide cream/lotion/gel | 21 (10.3%) |
| Topical retinoid cream/lotion/gel | 6 (2.9%) |
| Topical adapalene cream/lotion/gel | 18 (8.8%) |
| Antibiotic cream/lotion/gel | 24 (11.8%) |
| Combination cream/lotion/gel—combining above products | 29 (14.2%) |
| Antibiotics by mouth | 74 (36.3%) |
| Contraceptive pill | 39 (19.1%) |
| Cocyprindiol by mouth | 6 (2.9%) |
| Isotretinoin by mouth | 29 (14.2%) |

different treatments were being used at baseline (see table 2). Oral antibiotics were very common, with 80% having ever tried them and 36.5% taking them at baseline. A total of 167 (81.9%) participants completed follow-up at 6 weeks (see table 3).

Seventeen participants were recruited into the qualitative interview study through primary care. Most were female (76.5%) and the sample age ranged from 18 to 46 years (mean age=25.5 years). Most (10) were students. Six participants reported having one or more other long-standing conditions. Some of these, such as eczema, polycystic ovarian syndrome and fibromyalgia, were also discussed during the interviews in relation to some interview responses. These did not appear to have an impact on findings, however it was noted that there was some confusion between responding to the Skindex-16 when having both acne and eczema.

## Quantitative findings

### Internal consistency

All Cronbach alpha values at baseline (online supplemental table 1) were above 0.7 for all subscales (0.74–0.96), suggesting good internal consistency. These values were similar to previously published 'good' internal consistency values, ranging from 0.77 to 0.96, for the Acne-QoL domains.[17]

### Construct validity

Both the Skindex and CompAQ were moderately correlated with the patient global, with slightly lower point estimates for the CompAQ (table 4). The Spearman correlations for the total Skindex was 0.60 whereas the correlation for the CompAQ was 0.54.

When compared with the domains of the Acne-QoL (table 4), there were generally moderate to high correlation coefficients (0.54–0.81). These values were particularly high for the total scores (0.69–0.81), which may suggest that although the subdomains differ, overall the measures may all capture acne related QoL.

### Reliability

ICC values for both questionnaires and their subscales (online supplemental table 2) were high and above 0.7 (ranging from 0.88 to 0.97), indicating a very good reliability. These results were in line with previously published data for the Acne-QoL (ICC 0.84–0.90).[16]

### Responsiveness to change

Please see this data within the supplementary information (online supplemental information 1 and online supplemental table 3).

### Interpretability

Please see this data within the supplementary information (online supplemental information 2).

## Qualitative findings

From the cognitive think-aloud interviews, we identified two overarching themes from the data—'acceptability' and 'usability'—and within these various subthemes (see table 5). There was no consensus regarding preferences for the measures, with a number of both positive and negative comments specific to the measures and the questions within them.

### Acceptability

Participants commented on the acceptability of each of the three PROMs and this theme contained three subthemes, as set out below.

**Table 3** Characteristics at all time points

| How would you describe your acne at the moment? | Baseline (n=204) | 24–48 hours (n=204) | 6 weeks (n=167) |
|---|---|---|---|
| Clear | 15 (7.4%) | N/A | 10 (6.0%) |
| Almost clear | 50 (24.5%) | | 54 (32.5%) |
| Mild | 58 (28.4%) | | 43 (25.9%) |
| Moderate | 61 (29.9%) | | 46 (27.7%) |
| Severe | 17 (8.3%) | | 13 (7.8%) |
| Very severe | 3 (1.5%) | | 0 |
| Acne-QoL | | | |
| Symptoms (range 0–30) | 17.3 (6.9) | 17.4 (7.1) | 18.4 (7.2) |
| Role emotional (range 0–30) | 14.7 (9.2) | 14.4 (9.0) | 16.4 (9.2) |
| Role social (range 0–24) | 15.6 (7.7) | 15.9 (7.4) | 16.8 (7.4) |
| Self-perception (range 0–30) | 14.4 (10.0) | 14.5 (9.8) | 17.0 (9.3) |
| Skindex-16 | | | |
| Symptoms (range 0–24) | 8.3 (6.6) | 8.6 (7.1) | 8.4 (7.0) |
| Emotions (range 0–42) | 27.5 (12.1) | 26.8 (12.3) | 24.9 (12.7) |
| Functioning (range 0–30) | 11.5 (9.3) | 11.0 (9.3) | 10.0 (9.0) |
| Total (range 0–96) | 46.8 (24.9) | 45.7 (25.7) | 35.3 (28.6) |
| CompAQ | | | |
| Symptoms (range 0–32) | 19.4 (8.3) | 18.1 (9.1) | 18.4 (8.9) |
| Social (judgements by others) (range 0–32) | 12.1 (9.1) | 11.1 (9.4) | 10.6 (9.0) |
| Social interactions (range 0–32) | 10.6 (9.3) | 10.3 (9.3) | 9.4 (9.1) |
| Psychological/emotional (range 0–32) | 18.3 (9.1) | 17.1 (9.9) | 15.8 (9.3) |
| Treatment concerns (range 0–32) | 21.8 (7.6) | 20.8 (8.4) | 20.3 (8.2) |
| Total (range 0–160) | 82.1 (36.4) | 77.3 (39.8) | 74.6 (39.3) |

*Relatability*

Many questions were either explicitly stated, or implied as being either relatable or not relatable for a number of reasons. For example, this appeared to be linked to severity of the participant's acne, whereby relatability was higher with a high severity. For many questions, participants described them as not being relatable at that time, but when or if their acne was worse, they could imagine it being more relatable. Some participants described understanding that though something may not be relatable to them, it may be to other people. Another example of reasons for why participants thought something was relatable or not appeared to be individual preferences, such as how the person perceived the importance of personal appearance. Individual whole questionnaires were also at times described as being most relatable or not relatable

**Table 4** Spearman's correlation coefficient compared with patient global and Acne-QoL (n=204).

| | Spearman correlation with patient global | Acne-QoL self-perception | Acne-QoL role emotional | Acne-QoL role social | Acne QoL symptoms |
|---|---|---|---|---|---|
| Skindex-16 | | | | | |
| Symptoms | 0.52 | 0.61 | 0.61 | 0.60 | 0.57 |
| Emotions | 0.60 | 0.79 | 0.79 | 0.74 | 0.68 |
| Functioning | 0.47 | 0.79 | 0.68 | 0.81 | 0.58 |
| Total | 0.60 | 0.79 | 0.79 | 0.81 | 0.69 |
| CompAQ | | | | | |
| Symptoms | 0.62 | 0.65 | 0.62 | 0.60 | 0.67 |
| Social (judgements by others) | 0.43 | 0.61 | 0.60 | 0.69 | 0.57 |
| Social interactions | 0.34 | 0.64 | 0.61 | 0.75 | 0.54 |
| Psychological/emotional | 0.53 | 0.75 | 0.74 | 0.73 | 0.61 |
| Treatment concerns | 0.37 | 0.58 | 0.67 | 0.57 | 0.58 |
| Total | 0.54 | 0.75 | 0.76 | 0.79 | 0.69 |

Acne-QoL, Acne-specific Quality of Life; CompAQ, Comprehensive Acne Quality of Life Scale.

**Table 5** Qualitative themes and subthemes

| Overarching theme | Subtheme | Example quote |
|---|---|---|
| Acceptability | Relatability | 'That really does bother me actually; it's like – if I stop using the cream, then it will stop working. That's a big annoyance of mine, because with most medical conditions you have a treatment and it's cleared, but this one, it's very – even though it's quite a small condition and it's not physically – it's not going to end your life or anything, it does feel quite restrictive.' |
| | Influence of questionnaire and emotive reaction to wording | 'It's like a little bit shocking when you first read them, because they are instantly like – do you feel depressed, do you feel tired, whereas I think there's a lot of other ones that – they float around the subject, whereas I think this is probably more – in your face.' |
| | Need for other health concerns | 'I know it's difficult. So, yes, maybe a bit of a broader … question about other – other aspects of your health'. |
| Usability | Layout | 'I like the way that one looks. It's a bit more spaced out (slight laughter from both), all on one page, which is nice, because you don't feel like you've got to sit there and go through pages and pages. You can see a picture from the ticks and stuff, for me, looking at it and – whereas that one was kind of like – just went on and on, but actually if you condensed that, that would actually be really similar'. |
| | Question and response wording | 'I'm struggling a little bit to understand on that one. It's just the way it's worded ever so slightly. It's like it's asking for a positive but a negative at the same time, about how concerned were you that your medication products were working fast enough…' |
| | Redundant and similar questions | 'I'm trying to just answer each question individually, but in the back of my head, I'm thinking – oh, do I want to turnover and see what I put for one of the others, because it feels like the same (I: Okay) sort of question. But maybe – you probably can't tell me, but are they doing that because they're checking to see how consistent I am with my answers? I don't know?' |
| | Guidance notes | 'I liked the middle one because it had some information at the top and I felt like it was clear, although it was more to read, it was clear about what I was doing, which this one doesn't have, in the same way' |

to them. For example, one participant described the Skindex-16 as most relatable (data extracts are shown in table 5).

### Influence of questionnaire and emotive reaction to wording

Some participants described negative or positive emotive reactions when filling out the questionnaire(s), such as being shocked or pleased about the way questions were worded. This was also implied at various points throughout interviews. Some participants also described how the questionnaire and/or their emotive reaction to the questions may have influenced their answers to the questions. For example, when they thought a questionnaire or specific question was quite negative, they reported that they may have experienced a negative reaction and may have answered the question more extremely/negatively.

### Need for other health concerns

Some participants suggested that it was difficult to answer some questions due to other health concerns that they have but were unable to discuss when filling it in, such as eczema. This was at times described as a suggestion for

future changes; adding in a section where you can state other health concerns or talk about other health issues that they have alongside their acne.

### Usability

Four subthemes emerged that related to the usability of the PROMs and these are explained below.

### Layout

All PROMs at times were described as having a likeable or dislikeable layout and there were often mixed responses regarding specific parts. The layout of the Acne-QoL particularly seemed to evoke mixed responses. It appeared to be likeable at times, but also appeared to seem longer to due to consisting of three pages (although the number of questions is similar). Many participants thought that the bold keywords within the Skindex-16 were useful in highlighting the meaning of the question, however it was also suggested that more labels for the response numbers would have been useful in answering the questions. Participants discussed the grid layout of the CompAQ and while some liked this layout, others

described it as 'amateur' looking. At times this grid layout seemed to make filling the questionnaire harder, leading to accidentally missing questions.

### Question and response wording

Participants described the wording of questions as negative, personal to individuals, strong or more emotive. At times, the conciseness of the questions were commented on, such as being too ambiguous or vague to the situation it describes. The term 'never bothered' within the Skindex-16 seemed to be somewhat challenging and ambiguous, in that participants were unsure as to what it was specifically asking them to score. Participants debated whether the term 'never bothered' was asking them to answer according to whether they simply experienced the symptoms referred to in the question, or whether they experienced symptoms and felt specifically bothered by them. Moreover, response options, such as the words used within the Acne-QoL or numbers used in the Skindex-16 and CompAQ were described as both helpful and unhelpful. For example, at times, the word options were described as too subjective and interpretation could vary between people. Number options also appeared to be too vague and led to arbitrarily choosing a number. On the other hand, for example, the scale of various response options were described as helpful.

### Redundant and similar questions

Some questions within the same measure were described as similar and were often perceived as unnecessary or irritating. For example, a participant indicated that this made them wonder whether they were being tested in their answers and therefore went back to check what they had answered previously. The similarity in questions, however was also described as reassuring or helpful by some. Some questions were described as obvious and would therefore be unnecessary in a questionnaire.

### Guidance notes

Various notes on questionnaires were described as helpful or unhelpful. Instructions at the top of the CompAQ were often described as helpful, though others skipped these or did not read fully. Several participants also said they found the definition of acne on the CompAQ helpful in order to think about what the questions are asking. However, it had also been described as not inclusive enough due to not including scarring. The time period of the last 7 days on the Acne-QoL and Skindex-16 was often seen as frustrating because participants described not being able to answer how they generally feel (usually due to being interviewed on a good week). In other cases, this was described as helpful, because the specific instruction of the last 7 days made it easier to think of an answer in a shorter time period. On the other hand, with no instruction of a particular time period on the CompAQ, some participants were confused as to whether to think of a time period, how they currently feel or how they generally feel.

## DISCUSSION

This study explored the validity and acceptability for adult acne of the Skindex-16 and the CompAQ, which have not been validated previously for use in acne research. Both measures performed well in comparison to the validated Acne-QoL[16 17] and patient-reported global measure. Internal consistency and reliability were high and responsiveness to change was as anticipated for most subscales.

While the instruments showed high levels of correlation with Acne-QoL and with the patient global measure, it was clear that they do not all capture the same domains. In particular, the 'treatment concerns' domain in the CompAQ was a new domain that does not appear in the Skindex or Acne-QoL. Furthermore, although there were floor effects in some domains, some participants had reported their acne to be very mild or clear at present, which may have been the reason for scoring the lowest value throughout the measures.

Qualitatively, there were mixed views across the sample. Participants expressed both positive and negative comments about aspects of each measure, regarding acceptability and usability and their subthemes, as well as different questionnaire preferences (ie, no one questionnaire appeared superior or inferior across the whole sample). These findings may have been due to individual differences in what the participant most wants and needs, but again highlights that though all measures look at QoL for acne, their domains and their angle of looking at QoL slightly differ. This study used cognitive interviews to assess the PROMs after they have been created, however other studies have used cognitive interviews to develop acne PROMs.[18 33] For example, cognitive interviews were used in phase 2 of the development of the CompAQ,[18] whereby data also emerged based on response options, clarity of the items and redundant questions.

### Strengths

The mixed methods in this study allow for triangulated findings with both exploration and analyses. The rich findings from the qualitative interviews helped support the quantitative analyses which did not suggest one measure to be superior to the others. The sample size of this study, 204 quantitative participants, was in line with the recommendations of over 100 participants[23] and at least three times as many participants as the number of questionnaire items.[25] Seventeen qualitative interviews also was sufficient to achieve data saturation of the main themes. Moreover, both samples include a wide age range of adults, from 18 to 49, making the findings more generalisable to a wider adult acne population. Also within the quantitative sample, a variety of acne treatments had been tried or were currently being used, indicating a variety of different people with acne.

### Limitations

Though current and past treatments were indicated in the questionnaires, the reporting of acne severity in acne questionnaires, and also when described in interviews,

was subjective. There were also no qualitative interviewees recruited to the study through secondary care and many of the quantitative participants were recruited through primary care, which made generalisability (of quantitative findings) and transferability (of qualitative findings) to all severities of acne difficult. Nevertheless, a range of perceived severities were reported across questionnaires and even some of the interviewees discussed having been on treatment that is given in secondary care, such as isotretinoin.

It must also be considered that assessment of responsiveness to change and interpretability within this study were challenging. Floor/ceiling effects may be due to the many participants who reported that their acne was 'clear' or 'almost clear' (see table 3). Only 44 quantitative participants reported changes in treatment during the 6-week period between completing questionnaires. Since most patients had relatively stable acne over the study period, it is challenging to assess responsiveness to change over time. Responsiveness to change and other measurement properties may differ in a clinical trial setting, where the patient population is more well defined with regard to acne severity and location. Moreover, responsiveness to change requires the participant to assess their change from baseline, which may be subject to recall bias. We did suggest that participants take a photo at baseline to assist them with this assessment but we do not know how many actually did so.

The sample was predominantly female, which might limit generalisability. However, the prevalence of adult acne is higher in females[34] and our sample may therefore reflect this. Moreover the study was only undertaken in adults and therefore the results may not be generalisable to those under 18. Since acne is very prevalent in this population, further research would be needed to establish whether these results extend to this population.

## CONCLUSION

All PROMs are valid but qualitative data suggest that they may not meet exactly what all participants want, capturing different aspects of QoL. In further research studies and trials with an adult acne population, any PROM should be acceptable for use. However, it is important that researchers reflect on which PROM would be most suitable for their study population, considering the differences in domains of QoL measured within the PROMs, the location of the acne and the time frame over which the measures will be used. The future development of a new PROM covering more QoL domains, or further evaluation of novel PROMs[33] may also be beneficial.

**Acknowledgements** The research team acknowledges the support of the National Institute for Health Research Clinical Research Network (NIHR CRN). The authors thank the participants, GP practices, trusts and PPI contacts for their involvement in this study.

**Contributors** All authors have been included. SH and BS contributed equally to this paper. MS, BS, IM, AML, LM, PL and KT designed the study and contributed to the funding application and protocol. SH collected the data and carried out the qualitative analysis with input from IM, MS and BS. BS and JK analysed the quantitative data. SH and BS wrote the first draft of the paper with substantial contribution from IM and MS. All authors have read and contributed to the drafts of the manuscript and have approved the final version.

**Funding** This study was funded by the National Institute for Health Research (NIHR) School for Primary Care Research (project reference 369). The views expressed are those of the authors and not necessarily those of the NIHR or the Department of Health and Social Care.

**Competing interests** AML is an active member of ACORN (Acne Core Outcome Research Network) involved in establishing standardised and agreed core outcome measures for acne trials; coauthor and principle investigator for ompAQ; acted as Chief investigator for Research studies supported by Galderma and GSK pharmaceuticals; acted as clinical advisor over the last 5 years and provided unrestricted educational talks supported by an honorarium for Galderma, La Roche Posay, Origimm, Proctor and Gamble.

**Patient consent for publication** Not required.

**Provenance and peer review** Not commissioned; externally peer reviewed.

**Data availability statement** Data are available upon reasonable request.

**ORCID iDs**
Samantha Hornsey http://orcid.org/0000-0003-1521-248X
Paul Little http://orcid.org/0000-0003-3664-1873

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
