## [Reviewer comments · BMJ Open]

ARTICLE DETAILS

TITLE (PROVISIONAL)	Patient Reported Outcome Measures for Acne: a Mixed Methods Validation Study (Acne PROMS).
AUTHORS	Hornsey, Samantha; Stuart, Beth; Muller, Ingrid; Layton, Alison; Morrison, Leanne; King, Jamie; Thomas, Karen; Little, Paul; Santer, Miriam

VERSION 1 – REVIEW

REVIEWER	Pavel Chernyshov National Medical University, Ukraine
REVIEW RETURNED	04-Oct-2019

GENERAL COMMENTS	The manuscript is dedicated to a narrow question: validation of two PROs in acne patients. However introduction and its subsection “acne outcomes” are not related to study objectives. Subsection of methods section entitled “Patient and Public Involvement” may be presented as supplementary material or deleted. It is strange that authors have included into analysis one (one!) patient recruited by “community advertising”. What was the reason? How the diagnosis of acne was confirmed in this case? One patient, probably, did not influence on statistical analysis in this study but may negatively influence its methodology. I propose to exclude results of that patient and reanalyze data. Another methodological problem is comparison of authors’ data on Skindex-16 and CompAQ with previously published data on Acne-QoL. Because of different acne patient characteristics in different studies the results for Acne-QoL may also be not identical. It seems better to compare authors’ own data with previously published data in discussion section or otherwise each time and in each table clearly state that Acne-QoL data was adopted from other studies. It is better to present “Qualitative finding” results in numeric data instead of general phrases like “many questions”, “some participants”, “some questions”, “many participants” that are more proper for discussion section. It is important to discuss results of other studies in this field.
---

REVIEWER	Diane Thiboutot Penn State University United States I participated in the initial development of the CompAQ and the Acne-QoL instruments.
REVIEW RETURNED	08-Nov-2019

GENERAL COMMENTS	The authors are to be commended for their efforts to generate data on the validity, usability and acceptability of the relatively new CompAQ and the Skindex-16 as a patient reported outcome measures for acne. These data represent a contribution to the field that will assist international efforts to select core outcome measures to assess acne for both clinical practice and clinical trials. This mixed-methods study was conducted in a real world clinical setting as opposed to a clinical trial. The strengths of this paper are in the assessments of internal consistency and construct validity of the instruments in 204 adults seeking consultation or treatment for acne within the prior 6 months; the comparison of measurement properties with the validated Acne-QoL and the generation of qualitative data on usability and acceptability in 17 acne patients. There are weaknesses in the assessment of responsiveness to change and interpretability based on the study design, several of which are recognized by the authors. Only 44/204 participants in the quantitative portion of the study experienced a treatment change in the 6 weeks following baseline. A 6- week time frame to detect change in acne is short, even for most all treatments. As pointed out by the authors, the observation of floor and ceiling effects of the instruments most likely relates to the lack of uniformity of acne severity in the sample with many subjects reporting “clear” or “almost clear”. Would recommend moving these data on responsiveness to change and interpretability to supplemental information and more strongly state the limitations of the study design that preclude meaningful assessment of these 2 measurement properties in this study. As pointed out by the authors, with regard to the qualitative aspect of the study, a major stumbling point was the inclusion of patients that had other skin disease in addition to acne- that confounded the findings particularly with the Skindex-16. Minor points;  1. There may be a typo in the abstract where it states that “Both scales showed floor effects on some subdomains. Skindex-16 and CompAQ showed good evidence of construct validity when compared to Acne-QoL with Spearman’s correlation coefficients 0.54 to 0.81, and good responsiveness to change over 24 hours.” The reviewer wonders if it should read “6 weeks”. 2. Table 4 should be renamed as the current title reflects only the first data column 3. Greater clarity is needed for this sentence in the results of responsiveness to change. For example, “As expected the correlations were higher in those that reported a change in treatment during the 6 week period with all scales above the threshold of 0.7 set for AUROC for those reporting a change” 4. Other aspects that should be mentioned in the discussion:
--

	 • Emphasis that this study was conducted in adults and would not be generalizable to a large proportion of the acne population (less than 18 yrs.) • Measurement properties may differ in a clinical trial setting where the patient population is more well defined with regard to acne severity and location
--	---

REVIEWER	Catherine Welch University of Leicester, England.
REVIEW RETURNED	11-Dec-2019

GENERAL COMMENTS	This is an excellent paper. Interesting topic, clearly written and appropriate methods used. I have one minor suggestion. Please could you describe in more detail how "purposive sampling" is used to calculate the sample size of a qualitative study?
--

REVIEWER	Professor Stephen Walters University of Sheffield UK
REVIEW RETURNED	07-Jan-2020

GENERAL COMMENTS	Patient Reported Outcome Measures for Acne: a Mixed Methods Validation Study (Acne PROMS). Hornsey, Samantha*1; Stuart, Beth*1; Muller, Ingrid1; Layton, Alison M2,3; Morrison, Leanne1,4; King, Jamie1; Thomas, Karen5; Little, Paul1; Santer, Miriam1. All the tables and analyses need to be clear what the sample size was. There were 204 participants at baseline and possibly 166 participants responded to the PROMS at 6 -weeks although this is not clear. The manuscript would benefit from a participant flow diagram showing how many participants completed each of the four PROMS - the Acne-QoL, Skindex-16, CompAQ and Global rating at each of the three time points baseline 24 hours and 6 weeks. Since the paper is about acceptability it would also be helpful to see the individual item completion rates at baseline? Did all 204 participants to the quantitative survey complete all the items in Acne-QoL, Skindex-16, CompAQ and Global rating? I Table 1 . The number of items for each of the domains need to be described. For example the number of the 19 items in the Acne-QoL symptoms domain need to be reported along with the range of possible scores for this domain. For assessing the test -retest reliability of a questionnaire or scale then the papers by
---

	Walter S.D., Eliasziw M., Donner A. Sample Size and Optimal Designs for Reliability Studies. Statistics in Medicine 1998; 17: 101-110. Provide a justification for the sample size. Table 2 Report number and percentage of sample who are male. Table 3 Report the sample size for each outcome and domain at each time point. Table 3 add the % of the sample who are at the floor and ceiling score for each outcome and dimension. Table 4 add the sample size for each correlation. Supplementary Table 1 Responsiveness to change over six-week period. Add a table with the mean change in PROM score over time by the self-reported patient global assessment of change group – either one of the six -point categories or collapse the six-point change into no change, acne better or acne worse. Compare mean change scores with a one-way ANOVA or similar. Calculate the standardised response mean (SRM)for the change in PROM score for each domain. Jaeschke, R., Singer, J., Guyatt, G.H. (1989) Measurement of Health Status. Ascertaining the Minimal Clinically Important Difference. Controlled Clinical Trials, 10, 407-415. Use the patient global assessment at baseline and six-weeks rated clear to severe to classify patients as no change, 1 category better, 2 categories better etc or 1 category worse , 2 categories better Add a table with the mean change in PROM score over time by the self-reported patient global assessment of change group . Compare mean change scores with a one-way ANOVA or similar. Calculate the standardised response mean (SRM)for the change in PROM score for each domain. Table 5 report the sample size for each correlation. Add sample size for each row, number of items in each domain. The high values of Cronbach’s alpha e.g. 0.95 for the Functioning domain of Skindex-16 suggest some items could possibly be removed from the domain? What about reporting Cronbach’s alpha with each item omitted one at a time? Supplementary Table 2 Add sample size for each row, number of items in each domain. Add a 95% CI for the ICC.
--	--

REVIEWER	Sangchoon Jeon Yale University, United States
REVIEW RETURNED	24-Jan-2020

GENERAL COMMENTS	This study was systemically performed to evident the validation of the measurement. However, I like to clarify some statistics which may not be fully described.
--

	 1. For internal consistency, Cronbach's alphas might be computed with baseline data only. But a specific time point for Cronbach's alpha is not described. 2. In Table 5, I believe the responsiveness to change should be evaluated by correlation between CHANGED score of the measurement and the Global assessment of change. However, seems the authors computed the correlations with total scores at 6 weeks (even time point is not specified) Not with the change total scores. This need to be clarified. 3. In Table 5, AUROC could be over-rated due to skewed distribution of global assessment of change. For example, if most cases are improved, it artificially increase AUROC. Therefore, it would be good to present sensitivity/specificity, plot of ROC curve, or at least the proportions of improve/not improve at 6 weeks. 4. The authors stated limitation of generalizability due to the predominant female samples. Male participants might have different perception and sensitivity of response to the questionnaire. It would be good to know how the constructive validity is different after eliminating male participants. 5. Minor recommendations: Many abbreviations are not initialized with full words.
--	---

VERSION 1 – AUTHOR RESPONSE

Reviewer: 1

Reviewer Name: Pavel Chernyshov

Institution and Country: National Medical University, Ukraine Please state any competing interests or state 'None declared': None declared

Please leave your comments for the authors below

The manuscript is dedicated to a narrow question: validation of two PROs in acne patients. However introduction and its subsection “acne outcomes” are not related to study objectives.

Since the readership of the journal represents clinicians and academics who may not be directly involved in acne research, we felt it was important to set the background motivation for this paper.

Acne research is important to patients but the outcomes used are rarely consistent between trials and rarely use consistent outcome measures.

Subsection of methods section entitled “Patient and Public Involvement” may be presented as supplementary material or deleted. – This section is required by the journal so I’m afraid we cannot omit or move it.

It is strange that authors have included into analyzis one (one!) patient recruited by “community advertising”. What was the reason? How the diagnosis of acne was confirmed in this case? One patient, probably, did not influence on statistical analyzis in this study but may negatively influence its methodology. I propose to exclude results of that patient and reanalyze data. – Community advertising was a planned method of recruitment to the study. Although it did not prove popular, the participant was still eligible as per the protocol. A sensitivity analysis excluding this person does not show any differences in the overall results or inferences. However, given that the participant was eligible and completed the study we don’t feel it would be ethical to exclude the data.

Another methodological problem is comparison of authors’ data on Skindex-16 and CompAQ with previously published data on Acne-QoL. Because of different acne patient characteristics in different studies the results for Acne-QoL may also be not identical. It seems better to compare authors’ own data with previously published data in discussion section or otherwise each time and in each table clearly state that Acne-QoL data was adopted from other studies. – We agree. However as we noted in the Methods section, the copyright for Acne-QoL is held by a pharmaceutical company and the

permissions we were given for its use specified that we could only compare to published values. To do as the reviewer suggests would, unfortunately, violate the copyright agreement.

It is better to present “Qualitative finding” results in numeric data instead of general phrases like “many questions”, “some participants”, “some questions”, “many participants” that are more proper for discussion section.

It is important to discuss results of other studies in this field.

The use of numbers and percentages are not usually used in qualitative research and it is not recommended in thematic analysis. There is a full explanation on p21 of FAQs at www.psych.auckland.ac.nz/en/about/thematic-analysis

We have added discussion of other studies which used cognitive interview methods, in the discussion section.

Reviewer: 2

Reviewer Name: Diane Thiboutot

Institution and Country: Penn State University United States Please state any competing interests or state ‘None declared’: I participated in the initial development of the CompAQ and the Acne-QoL instruments.

Please leave your comments for the authors below

The authors are to be commended for their efforts to generate data on the validity, usability and acceptability of the relatively new CompAQ and the Skindex-16 as a patient reported outcome measures for acne. These data represent a contribution to the field that will assist international efforts to select core outcome measures to assess acne for both clinical practice and clinical trials. This mixed-methods study was conducted in a real world clinical setting as opposed to a clinical trial.

The strengths of this paper are in the assessments of internal consistency and construct validity of the instruments in 204 adults seeking consultation or treatment for acne within the prior 6 months; the comparison of measurement properties with the validated Acne-QoL and the generation of qualitative data on usability and acceptability in 17 acne patients.

There are weaknesses in the assessment of responsiveness to change and interpretability based on the study design, several of which are recognized by the authors. Only 44/204 participants in the quantitative portion of the study experienced a treatment change in the 6 weeks following baseline. A 6- week time frame to detect change in acne is short, even for most all treatments.

As pointed out by the authors, the observation of floor and ceiling effects of the instruments most likely relates to the lack of uniformity of acne severity in the sample with many subjects reporting “clear” or “almost clear”.

Would recommend moving these data on responsiveness to change and interpretability to supplemental information and more strongly state the limitations of the study design that preclude meaningful assessment of these 2 measurement properties in this study.

Thank you for this suggestion. We have done this.

As pointed out by the authors, with regard to the qualitative aspect of the study, a major stumbling point was the inclusion of patients that had other skin disease in addition to acne- that confounded the findings particularly with the Skindex-16.

Minor points;

1. There may be a typo in the abstract where it states that “Both scales showed floor effects on some subdomains. Skindex-16 and CompAQ showed good evidence of construct validity when compared to Acne-QoL with Spearman’s correlation coefficients 0.54 to 0.81, and good responsiveness to change over 24 hours.” The reviewer wonders if it should read “6 weeks”. Thank you. This should actually have said “repeatability over 24 hours”. We have amended this.

2. Table 4 should be renamed as the current title reflects only the first data column Amended to include “and Acne QoL”

3. Greater clarity is needed for this sentence in the results of responsiveness to change. For example, "As expected the correlations were higher in those that reported a change in treatment during the 6 week period with all scales above the threshold of 0.7 set for AUROC for those reporting a change"

Amended as suggested

4. Other aspects that should be mentioned in the discussion:

- Emphasis that this study was conducted in adults and would not be generalizable to a large proportion of the acne population (less than 18 yrs.)

Thanks for the suggestion. We have added the following text to the limitations section : "Moreover the study was only undertaken in adults and therefore the results may not be generalisable to those under 18. Since acne is very prevalent in this population, further research would be needed to establish whether these results extend to this population."

- Measurement properties may differ in a clinical trial setting where the patient population is more well defined with regard to acne severity and location

We agree and have added this to the limitations section.

Reviewer: 3

Reviewer Name: Catherine Welch

Institution and Country: University of Leicester, England.

Please state any competing interests or state 'None declared': None declared

Please leave your comments for the authors below This is an excellent paper. Interesting topic, clearly written and appropriate methods used. I have one minor suggestion. Please could you describe in more detail how "purposive sampling" is used to calculate the sample size of a qualitative study?

Thank you for the comment. We have reworded and added to this text in the 'sample size' section on page 6 to make it clearer: The sentence was changed from "The qualitative study aimed to include 15 to 20 participants using purposive sampling" to " The qualitative study aimed to include 15 to 20 participants where data saturation was expected. Purposive sampling was used to select study participants to ensure a wide range of views and experiences were represented. For example, trying to include more male participants towards the end of the interview phase, because the sample consisted of more females."

Reviewer: 4

Reviewer Name: Professor Stephen Walters Institution and Country: University of Sheffield UK

Please state any competing interests or state 'None declared': None

Please leave your comments for the authors below Patient Reported Outcome Measures for Acne: a Mixed Methods Validation Study (Acne PROMS).

Hornsey, Samantha*1; Stuart, Beth*1; Muller, Ingrid1; Layton, Alison M2,3; Morrison, Leanne1,4; King, Jamie1; Thomas, Karen5; Little, Paul1; Santer, Miriam1.

All the tables and analyses need to be clear what the sample size was.

There were 204 participants at baseline and possibly 166 participants responded to the PROMS at 6 - weeks although this is not clear.

The manuscript would benefit from a participant flow diagram showing how many participants completed each of the four PROMS - the Acne-QoL, Skindex-16, CompAQ and Global rating at each of the three time points baseline 24 hours and 6 weeks.

Thank you. We have clarified this adding the following text: "At baseline and 24 hours, 204 participants completed all the PROMS and at 6 weeks, 167 participants completed the PROMS. No questionnaires were returned with uncompleted measure. All participants who returned the booklets to us completed all the question items on all the included PROMS." We have clarified this in the results section (page 8) of the text but since the flow diagram is effectively the same for all the measures, we haven't added a flow diagram.

Since the paper is about acceptability it would also be helpful to see the individual item completion rates at baseline? Did all 204 participants to the quantitative survey complete all the items in Acne-QoL, Skindex-16, CompAQ and Global rating? I

Yes, all participants completed all the questions required to score the instruments and we have added the text above to address this. We defined acceptability more broadly, using the qualitative think aloud interviews as our method of assessing this rather than a quantitative approach.

Table 1 . The number of items for each of the domains need to be described.

For example the number of the 19 items in the Acne-QoL symptoms domain need to be reported along with the range of possible scores for this domain.

Thank you. Amended. The range of scores are the same for each domain, within each measure, so we have made this clearer within the description column for each measure.

For assessing the test -retest reliability of a questionnaire or scale then the papers by Walter S.D., Eliasziw M., Donner A. Sample Size and Optimal Designs for Reliability Studies. *Statistics in Medicine* 1998; 17: 101-110.

Provide a justification for the sample size.

The overall sample size is given in the method section. It was not specifically calculated just to look at test-retest reliability.

Table 2

Report number and percentage of sample who are male.

Amended

Table 3

Report the sample size for each outcome and domain at each time point.

Amended – since all participants who completed the measures answered all questions on the domains, we have just given the overall sample size at each timepoint as it applies to all items in that column.

Table 3 add the % of the sample who are at the floor and ceiling score for each outcome and dimension.

Thank you for the suggestion. However, another reviewer suggested moving the data on floor and ceiling effects to the supplementary material. We have made this change and therefore felt that adding this data to Table 3 would be confusing. If the editor feels this additional data is essential, we could add another table to the Supplementary material to complement the text that is now presented there.

Table 4 add the sample size for each correlation.

Thank you. The sample size is the same for all correlations so we have added this into the title of the table

Responsiveness to change over six-week period.

Add a table with the mean change in PROM score over time by the self-reported patient global assessment of change group – either one of the six -point categories or collapse the six-point change into no change, acne better or acne worse.

Compare mean change scores with a one-way ANOVA or similar. Calculate the standardised response mean (SRM) for the change in PROM score for each domain.

Jaeschke, R., Singer, J., Guyatt, G.H. (1989) Measurement of Health Status. Ascertaining the Minimal Clinically Important Difference. *Controlled Clinical Trials*, 10, 407-415.

Whilst this is an approach often used to assess responsiveness to change, it is not the approach recommended by the COSMIN guidelines, which we have followed throughout this assessment.

Indeed the guidelines stated that this approach was considered by COSMIN to be “inappropriate” (see for example Mokkink et al 2010

<https://bmcmmedresmethodol.biomedcentral.com/articles/10.1186/1471-2288-10-22>) Only approaches which assess responsiveness using an AUROC approach are rated by the COSMIN as very good and therefore we have followed this approach. There are some limitations inherent in the assessment of

responsiveness to change in this population, as raised by another reviewer, and we have therefore moved this material to the Supplementary information rather than including it in the main paper.

Use the patient global assessment at baseline and six-weeks rated clear to severe to classify patients as no change, 1 category better, 2 categories better etc or 1 category worse, 2 categories better. Add a table with the mean change in PROM score over time by the self-reported patient global assessment of change group.

Compare mean change scores with a one-way ANOVA or similar. Calculate the standardised response mean (SRM) for the change in PROM score for each domain.

As above, this approach is not the one recommended by the COSMIN guidelines which we followed.

Table 5 report the sample size for each correlation.

Add sample size for each row, number of items in each domain.

Amended – the sample size for each row is the same as the sample size for that column. The number of items in each domain is reported in Table 1.

The high values of Cronbach's alpha e.g. 0.95 for the Functioning domain of Skindex-16 suggest some items could possibly be removed from the domain? What about reporting Cronbach's alpha with each item omitted one at a time?

This may well be true and would make an interesting additional study. However, the purpose of this paper was not to develop a new instrument, nor to suggest items which might be removed. Instead we aimed to assess the measurement properties of the instruments as they had been developed.

Supplementary Table 2

Add sample size for each row, number of items in each domain. Add a 95% CI for the ICC.

Thank you for the suggestion. As above, the sample size was the same throughout so we have added (n=204) to the title of the table. We have also added the 95% confidence intervals for the ICC.

Reviewer: 5

Reviewer Name: Sangchoon Jeon

Institution and Country: Yale University, United States Please state any competing interests or state 'None declared': None declared Please leave your comments for the authors below This study was systemically performed to evident the validation of the measurement. However, I like to clarify some statistics which may not be fully described.

1. For internal consistency, Cronbach's alphas might be computed with baseline data only. But a specific time point for Cronbach's alpha is not described. Apologies – this was at baseline. We have clarified this in the text.
2. In Table 5, I believe the responsiveness to change should be evaluated by correlation between CHANGHED score of the measurement and the Global assessment of change. However, seems the authors computed the correlations with total scores at 6 weeks (even time point is not specified) Not with the change total scores. This need to be clarified. Apologies that this was not clear. We did indeed do as suggested by the reviewer and compared the global assessment of change with the change scores for each measurement between baseline and 6 weeks. We have made this clearer in the methods by adding the text “their global assessment of change would correlate with their the change in their PROM scores between baseline and 6 weeks”
3. In Table 5, AUROC could be over-rated due to skewed distribution of global assessment of change. For example, if most cases are improved, it artificially increase AUROC. Therefore, it would be good to present sensitivity/specificity, plot of ROC curve, or at least the proportions of improve/not improve at 6 weeks. We agree and this is the reason that we also provided the correlations alongside the AUROC. Rather than making more of this analysis with additional statistics, we feel it is important not

to over interpret these results. Given the limitations of this analysis, we have followed Reviewer 2's suggestion to move this to the supplementary material.

4. The authors stated limitation of generalizability due to the predominant female samples. Male participants might have different perception and sensitivity of response to the questionnaire. It would be good to know how the constructive validity is different after eliminating male participants.

Yes, the results are unchanged. This may, of course, be because there were relatively few male participants in the sample so removing them has little impact.

5. Minor recommendations: Many abbreviations are not initialized with full words.
Amended

VERSION 2 – REVIEW

REVIEWER	Diane Thiboutot Penn State University I participated in the development of the Acne-QoL and the COMPAQ-AQ
REVIEW RETURNED	16-Jul-2020

GENERAL COMMENTS	Very nice work
----------------

REVIEWER	Sangchoon Jeon Yale University
REVIEW RETURNED	17-Jun-2020

GENERAL COMMENTS	All statistical approaches are appropriate for research aims. The authors may discuss the possibility of recall bias on the self-reported change score. This may lower the correlations between the self-reported change score and the actual difference in PROM over time (in Table 3). Not clear if 123 (=167 - 44) participants reported no change in the self-report also had zero delta score (score at 6wks - baseline) on PROM. If they reported zero on the self-reported score only but had different scores in PROM over time, then there is no reason to exclude them from the analysis.
---

VERSION 2 – AUTHOR RESPONSE

All statistical approaches are appropriate for research aims. The authors may discuss the possibility of recall bias on the self-reported change score. This may lower the correlations between the self-reported change score and the actual difference in PROM over time (in Table 3). Not clear if 123 (=167 - 44) participants reported no change in the self-report also had zero delta score (score at 6wks - baseline) on PROM. If they reported zero on the self-reported score only but had different scores in PROM over time, then there is no reason to exclude them from the analysis.

Thank you for these comments. We agree about the possibility of recall bias – we did try to minimise this by asking participants to take a photo at baseline but we don't know how many did this. We have added a sentence to this effect in the Discussion.

With respect to the responsiveness to change over time reported in Table 3, we agree with the reviewer and have not excluded any participants from the analysis. Rather we have divided the sample into two groups. Firstly we looked at the whole cohort who provided 6 week follow up

measures (n=167) and have presented these findings. Then we looked at those who had a change in their acne treatment, as we expected that these participants might have experienced larger changes compared to those who were stable on treatment and this does seem to have been the case (albeit in a small sample of n=44).